# Persistent Cyanobacteria Blooms in Artificial Water Bodies—An Effect of Environmental Conditions or the Result of Anthropogenic Change

**DOI:** 10.3390/ijerph19126990

**Published:** 2022-06-07

**Authors:** Paulina Nowicka-Krawczyk, Joanna Żelazna-Wieczorek, Izabela Skrobek, Maciej Ziułkiewicz, Michał Adamski, Ariel Kaminski, Paweł Żmudzki

**Affiliations:** 1Department of Algology and Mycology, Faculty of Biology and Environmental Protection, University of Lodz, Banacha 12/16, 90-237 Lodz, Poland; joanna.zelazna@biol.uni.lodz.pl (J.Ż.-W.); izabela.skrobek@edu.uni.lodz.pl (I.S.); 2Department of Geology and Geomorphology, Faculty of Geographical Sciences, University of Lodz, Narutowicza 88, 90-139 Lodz, Poland; maciej.ziulkiewicz@geo.uni.lodz.pl; 3W. Szafer Institute of Botany, Polish Academy of Sciences, Lubicz 46, 31-512 Cracow, Poland; m.adamski@botany.pl; 4Laboratory of Metabolomics, Faculty of Biochemistry, Biophysics and Biotechnology, Jagiellonian University, Gronostajowa 7, 30-387 Cracow, Poland; ariel.kaminski@uj.edu.pl; 5Department of Medicinal Chemistry, Faculty of Pharmacy, Jagiellonian University Medical College, Medyczna 9, 30-688 Cracow, Poland; pawel.zmudzki@uj.edu.pl

**Keywords:** artificial reservoir, CyanoHABs, eutrophication, global warming, human impact

## Abstract

Algal blooms are an emerging problem. The massive development of phytoplankton is driven partly by the anthropogenic eutrophication of aquatic ecosystems and the expansion of toxic cyanobacteria in planktonic communities in temperate climate zones by the continual increase in global temperature. Cyanobacterial harmful algal blooms (CyanoHABs) not only disturb the ecological balance of the ecosystem, but they also prevent the use of waterbodies by humans. This study examines the cause of an unusual, persistent bloom in a recreational, flow-through reservoir; the findings emphasize the role played by the river supplying the reservoir in the formation of its massive cyanobacterial bloom. Comprehensive ecosystem-based environmental studies were performed, including climate change investigation, hydrochemical analysis, and bio-assessment of the ecological state of the river/reservoir, together with monitoring the cyanobacteria content of phytoplankton. Our findings show that the persistent and dominant biomass of *Microcystis* was related to the N/P ratio, while the presence of *Aphanizomenon* and *Dolichospermum* was associated with the high-temperature end electric conductivity of water. Together with the increase in global temperature, the massive and persistent cyanobacterial bloom appears to be maintained by the inflow of biogenic compounds carried by the river and the high electric conductivity of water. Even at the beginning of the phenomenon, the reservoir water already contained cyanobacterial toxins, which excluded its recreational use for about half the year.

## 1. Introduction

Both the natural and artificial water reservoirs in urban and suburban areas are of great natural and socio-economic importance, with enduring value as recreational areas for the local community [1], and have been an integral part of the progressive development of civilization. The growing human impact on water reservoirs observed over the course of civilization has resulted in significant disturbances to ecosystem services, their safe use, and ecological sustainability [2]. The increase in trophic conditions caused by tourism, or by the inflow of nutrients from agricultural fields or pre-treated municipal wastewater, contributes to rapid, anthropogenic-based eutrophication [3,4].

One of the most serious consequences of eutrophication is the massive development of phytoplankton, which modifies the natural environmental conditions in aquatic ecosystems [5,6,7]. Such harmful algal blooms (HABs), composed of various types of algae, may occur in many types of ecosystems experiencing over-enrichment with nutrients [8]. Among the groups of bloom-forming algae, the greatest attention is paid to the cyanobacteria due to their ability to produce toxic cyanotoxins as products of their secondary metabolism [9,10,11]. The most common cyanobacteria genera producing cyanotoxins in phytoplankton are the following: *Microcystis*, *Aphanizomenon*, *Dolichospermum* (formerly planktonic *Anabaena*), *Cylindrospermopsis*, *Anabaenopsis*, *Planktothrix*, and *Nodularia* [12,13].

In reservoirs, acting as sources of drinking water, bloom formation has particular significance, as such contamination prevents their use [8]. The development of cyanobacteria in plankton is typically manifested by the formation of an “unsightly” floating blue-green biomass, a change in the color of the water, and an unpleasant odor. In addition to the loss of their ability to provide potable water, reservoirs also lose their recreational value, as the presence of cyanobacteria results in a ban on swimming, which also affects entrepreneurs through lowered tourism. However, the more important consequences of CyanoHABs concern public health since the use of water bodies during the presence of blooms carries the risk of being exposed to the harmful effects of toxins. These substances can exert an effect via direct exposure to toxic cyanobacteria cells, swallowing the water in which they are present, or just inhaling them with drops of water; direct contact with cyanotoxins may have negative effects on the health of both humans and animals, and can even be fatal [12,14,15].

Many artificial reservoirs have been created for recreational use by widening the riverbed and slowing down the river flow. The river ensures a constant supply of water, as well as constant access to the water basin by the construction of a small dam, even during the warm, dry summer. However, such construction disturbs the natural conditions of river ecosystems and may also negatively affect the communities of lenthic communities developing in reservoirs [16].

Global climate warming has been accompanied by an increase in the occurrence of cyanobacterial blooms in water bodies, and the resulting blooms are more burdensome in both freshwater and marine environments [3,17,18]. Cyanobacteria gain an advantage over other algae in phytoplankton blooms when the water temperature exceeds 20 °C [13]. Additionally, the rate of primary production in phytoplankton is also accelerated by high concentrations of biogenic compounds emitted by anthropogenic activity [3,9,18]. Therefore, increasing hydrobiological research is addressing the causes and effects of the appearance of blooms and, more importantly, the environmental exposure/risk assessment of CyanoHABs [5,10,19].

Due to the number of problematic issues arising from the blooms of recreational reservoirs, the present study focuses on one such site. The aim of the present investigation is to determine the causes and effects of a long-lasting cyanobacterial bloom in an artificial, flow-through, relatively young reservoir. The research determined whether natural forces such as global warming and human impact on the reservoirs are responsible for this unusually massive blooming phenomenon. A comprehensive hydrochemical and hydrobiological assessment was performed of the ecological status of the reservoir and of the river supplying the water basin before the bloom; this was followed by hydrochemical analysis and biomonitoring of the cyanobacterial bloom for half of the year. The findings provide an insight into the role of the supplying river in the formation of CyanoHAB.

## 2. Materials and Methods

### 2.1. Study Area

The Stryków artificial reservoir was created in 1991 for water sports and recreational purposes. It is located along 38 km of the Moszczenica river in the Łódź Metropolitan Area (Central Poland, Europe). The river course was slowed by a 110-m dam equipped with an overflow and discharge tower with a fixed upper overflow and adjustable bottom outlets. The reservoir covers an area of 12.6 ha, the maximum depth is 2.5 m, and the total capacity of the reservoir is 220,640 m^3^. As the reservoir is not emptied for the winter season, the water continually fills the reservoir throughout the year [20]. Recently, a few small aerators were set up at the surface of the reservoir to improve the water quality. The reservoir catchment area is bounded by the slopes of the Wzniesienia Łódzkie Landscape Park with 5.9–7.2% high differences in the ground [21]. The geology of the catchment originates from older glaciations, of which the youngest—the Warciański glaciation, left this area with thick fluvioglacial sediments and extensive zones of glacitectonic disturbances. The elevation of the area above the western, northern, and eastern adjacent regions contributes to higher precipitation and increased supply of atmospheric water. The amount of water runoff is 4–5 L/s km^2^ with a 70% share of underground infiltration [22].

This attractive landscape area is subject to strong human pressure as follows: many households and single-family houses have been constructed and the communication with the water supply infrastructure was developed. Moreover, a number of logistic centers have been constructed in this area as part of the development of the nearby junction of the A-1 and A-2 highways. Apart from a few isolated forest areas under reserve protection, the surroundings of the catchment are dominated by agricultural fields and orchards with wastelands.

For the purpose of this study, the following three sampling sites were selected: two at the Moszczenica river channel, one before the reservoir basin (site 1) and one after (site 2), and one at the reservoir (Figure 1).

### 2.2. Environmental Background

#### 2.2.1. Climate and Hydrochemical Data Analyses

The climate conditions with the greatest impact on the formation of cyanobacterial blooms were investigated during the study season (from 1 April 2019 to 30 November 2019) and over the previous 20 years, from January 1999 to December 2019. The distribution of mean daily (1 April 2019–30 November 2019) and monthly (January 1999–December 2019) temperatures were analyzed, together with the total daily/monthly precipitation. The meteorological data were obtained from the closest weather station of the Polish Institute of Meteorology and Water Management for Łódź–Lublinek.

To detect the changes in the hydrochemical conditions of the reservoir, physical and chemical water surveys were carried out all research sites, both in the period preceding the bloom formation (diurnal, hourly measurements from the 7 May 2019; *n* = 24 per site), and from the reservoir during its occurrence (once per field survey, from April to November 2019; *n* = 10). Water temperature (W-Temp), pH, electrical conductivity of water (EC), and dissolved oxygen (Diss-O_2_) were measured directly in situ, in the river channel and reservoir. All measurements were taken at a depth of 0.5 m using the following field research equipment: pH meter (Elmetron CP-401), conductometer (Elmetron CC-401), and oxygen meter (Elmetron CO-401). Nitrate (N-NO_3_), orthophosphate (P-PO_4_), and ammonium ions (N-NH_4_) concentrations were determined by spectroscopy, using a Marcel S.330 for samples from the 7 May and a Spectoquant Pharo 100 for those collected during the bloom. The analysis of total nitrogen (TN) and total phosphorus (TP) was performed by CFA (continuous flow analysis) with spectrophotometric detection. The statistical significance of the changes in hydrochemical conditions of river/reservoir ecosystem was calculated by Kruskal–Wallis test by ranks using Statistica ver. 13.3 software (StatSoft, Cracow, Poland). The analyses were performed in the Laboratory of the Department of Geology and Geomorphology at the Faculty of Geographical Sciences, University of Łódź (UniLodz), the Laboratory of Computer and Analytical Techniques, and the Department of Algology and Mycology at the Faculty of Biology and Environmental Protection, UniLodz, and in the Laboratory of the Group Sewage Treatment Plant in Łódź.

#### 2.2.2. Ecological Status Bioassessment

The biological assessment of the ecological status of the Moszczenica river and Stryków reservoir was carried out on the basis of benthic diatom communities collected from all three sampling sites before the bloom in spring 2019, and during the bloom in the reservoir. Benthic diatom samples were subjected to the standard chemical procedure according to [23]. The diatom communities were subjected to qualitative and quantitative analysis, and the diatom bioassessment was performed using the following diatom indices of trophic and saprobic conditions: IPS (Specific Pollution Sensitivity Index), IBD (Biological Diatom Index), GDI (Generic Diatoms Index), EPI-D (Eutrophication/Pollution Index Diatom-based), and the TDI (Trophic Diatom Index). The indices were calculated using OMNIDIA ver. 6.0.8 software (IRSTEA, Bordeaux, France) and interpreted according to [24]. Moreover, the autecology of dominant taxa was considered and interpreted as follows [25].

### 2.3. Investigation of Cyanobacterial Bloom

Phytoplankton samples were collected at the same time of day (11:00 am), monthly or at two-week intervals during heavy-bloom, using a KC-Denmark phytoplankton net with a mesh size of 0.25 μm. Each of the collected samples (*n* = 11) was immediately transported to the laboratory of the Department of Algology and Mycology, University of Lodz for further analysis.

The qualitative analysis was carried out on unfixed material, using the Nikon Eclipse 50i light microscope (Precoptic Co., Warsaw, Poland), under 400× and 600× magnification, using the following taxonomic references: [26,27,28,29]. Microphotographs were taken of cyanobacteria using an Opta-Tech HDMI digital camera (Opta-Tech, Warsaw, Poland).

For the quantitative analysis, the biomass of cyanobacteria taxa was calculated based on the ratio of the mean volume of colonies/trichomes of a particular taxa and its number per ml of sample. The abundance of taxa was determined by the drop method [30] using standard microscopic slides and the following equation:$${\mathrm{Lx}}_{i}=\frac{\mathrm{Ni}\times \mathrm{Pp}}{\mathrm{Pi}\times V\times Z}$$
where Lx_i_ is the number of individual taxa per 1 cm^3^ of sample, Ni is the number of colonies/trichomes counted, Pp is the total area of the microscopic slide, Pi is the area used for taxa counting, V is the volume of the drop, and Z is the concentration factor, i.e., the ratio of the volume of water flowing through the planktonic net to the final, concentrated volume of the sample.

The presence of the cyanobacterial bloom was estimated on the basis of total biomass of cyanobacteria in each sample and the threshold biomass value (B ≥ 3,000,000 μm^3^/mL) after [31].

Finally, the collected samples were fixed with 2% formaldehyde and deposited in the collection of the Department of Algology and Mycology, University of Lodz, Poland.

### 2.4. Multivariate Data Analysis

To summarize the variation in species composition and biomass and to interpret this summary with the help of the best-fitting subset of hydrochemical parameters, multivariate data analyses were performed. As the response data were compositional with a 1.3 SD gradient; the following unconstrained linear method was chosen: a principal component analysis (PCA) with supplementary variables. For the response data, centering and standardization by species were applied without any data transformation. The analysis was preceded by an interactive-forward-selection test to select the best-fitting hydrochemical factors for interpreting the summary in species composition. Factors that explained ≥10% of variation in species composition were chosen for PCA. Multivariate data analyses were performed using Canoco ver. 5.0 Software for Windows (Microcomputer Power, Ithaca, NY, USA).

### 2.5. Investigation of Cyanobacterial Toxins

The presence of cyanobacterial toxins in water was detected in the second half of June and the first half of July when the environmental conditions are most favorable for the development of cyanobacteria and there is the greatest interest in using aquatic ecosystems for recreation.

To determine the presence and concentration of cylindrospermopsin (CYN), microcystin LR (MC-LR), and anatoxin-a (ANTX-a), all of the samples were analyzed on a Shimadzu Nexera-I LC-2040C 3D Plus Ultra High Performance Liquid Chromatograph (UHPLC) (Shimadzu UK Ltd., Buckinghamshire, UK) [32], with the following detection limits: 18 ng/mL for CYN, 15 ng/mL for MC-LR and 12 ng/mL for ANTX-a. The gradient mobile phase consisted of water/acetonitrile (both acidified with 0.05% trifluoroacetic acid), where the organic phase increased from 2% to 90% over 15 min at a flow rate of 0.75 mL/min. Samples were separated on a Gemini^®^ NX-C18 Column (110 Å, 3.0 μm, 150 mm × 4.6 mm) (Phenomenex, Warsaw, Poland) maintained at 40 °C. Autosampler cooler temperature was 4 °C, and the PDA cell temperature was 40 °C. Toxins were identified by comparing the retention time and UV-spectra determined for commercial standards and quantified by absorbance at 227, 239, and 261 nm, for ANTX-a, MC-LR, and CYN, respectively. A multilevel calibration curve was obtained using commercial standards (from 0.01 to 10.00 μg/mL). The presence of the toxin in samples was confirmed using an ultra-performance liquid chromatography tandem-mass spectrometer (UPLC-MS/MS) coupled with a Waters TQD mass spectrometer (electrospray ionization mode ESI-tandem quadrupole) (Waters Corporation, Milford, MA, USA) [33,34].

## 3. Results

### 3.1. Environmental Background

#### 3.1.1. Climate and Hydrochemical Data Analyses

Significant changes in air temperature and precipitation were observed over the twenty-year data as follows: the mean value of the monthly mean air temperature of the entire research season in 2019 was 1.5 °C higher (14.3 °C) than the same period in 1999 (12.8 °C), while the mean monthly maximum increased by 2.5 °C, from 20.7 °C in 1999 to 23.2 °C in 2019 (Figure 2 and Figure 3). The highest temperature recorded in 1999 was 32.2 °C, noted on the 5 July, while twenty years later, in 2019, it reached 36.2 °C on the 30 June. The period for temperatures remaining above 18 °C also extended as follows: in 1999, this monthly average was noted only in July, while in 2019, it lasted for three months, i.e., from June to August. The linear regression of the mean monthly air temperatures over the 20-year period shows a gradual increase—the determined slope factor (a) is 0.0053 (Figure 3).

The mean monthly sum of precipitation was 31 mm in the 2019 research season and 51.7 mm in the parallel period of 1999. Moreover, the total sum of precipitation in the research season was 247.7 mm, giving only 60% of the total precipitation in 1999 (413.3 mm), even though the catchment area is characterized by higher precipitation than adjacent regions (see the following: Section 2.1). In total, there were 170 rainy days throughout 1999, but only 158 days 20 years later. The monthly sum of precipitation also demonstrated a downward trend over the last 20 years as follows: the determined slope factor (a) is 0.0145 (Figure 3). During the research season in 2019, no violent weather events or long-lasting precipitation were recorded.

At the time proceeding the formation of the bloom, the reservoir demonstrated a higher water temperature and pH than in the Moszczenica river (Figure 4A,B). In addition, the upper section of the river was characterized by higher electrolytic conductivity than the reservoir and supplied the reservoir with high loads of nitrates and orthophosphates (Figure 4C–F). Furthermore, the river section before the reservoir was found to have more than 1.5 times higher total nitrogen load than the section behind the reservoir (Figure 4G), while the influent river demonstrated a lower daily maximum total phosphorus content than the section behind the reservoir (Figure 4H). Both these nitrate and total phosphorus levels exceed the accepted values for surface water quality in Poland, according to the Regulation of the Ministry of Marine Economy and Inland Navigation in Poland [35]. Based on the hydrochemical conditions, i.e., both the high inflow of biogenic compounds carried by the Moszczenica river to the Stryków reservoir and the high amount of total phosphorus recorded next to the reservoir, indicates that the reservoir undergoes strong eutrophication at the beginning of the summer season.

The water temperature in the reservoir during the research season was 21.8 ± 2.7 °C, with the exception of 7 May and 11 July, when it was lower by 4.5 and 2 °C, respectively; in addition, it dropped below 10 °C after the 19 October (Table 1). The water in the reservoir was always alkaline. During the formation of the cyanobacterial bloom, an increase in dissolved oxygen concentration was recorded; however, in July, the concentration was the lowest of the summer season. The conductivity of water was low, with a mean value of 259 μS/cm (SD = 36), and this value tended to decrease over the course of the bloom. The N/P ratio was >17 in May and early June, following which it decreased significantly to ≤8.0 in the second half of June, which was caused by a significant reduction in nitrate concentration. Furthermore, at the beginning of July and at the turn of August–September, the nutrient ratio increased nine and five times, respectively, compared to the previous periods (Table 1).

#### 3.1.2. Ecological Status Bioassessment

The analysis of benthic diatom communities and the bioassessment of ecological status in spring 2019, i.e., before the vegetation season, showed the following: (1) in the upper section of the Moszczenica river, eutrophic conditions and moderate to poor ecological status predominated, with following dominants *Staurosira venter* (Ehrenberg) Cleve and Moeller (18.3%), *Staurosirella mutabilis* (W. Smith) E. Morales and Van de Vijver (7.8%), *Stephanodiscus hantzschii* Grunow in Cleve and Grunow (7.4%) and *Navicula rhynchotella* (6.7%) Lange-Bertalot; (2) in contrast, for the Stryków reservoir, hypertrophic conditions and poor to bad ecological status were noted, with dominants as *Stephanodiscus hantzschii*, *Cyclostephanos dubius* (Fricke) Round, *Aulacoseira granulata* (Ehr.) Simonsen, *Navicula tripunctata* (OF Müller) Bory, *Fragilaria vaucheriae* (Kützing) Petersen and with the 73% participation of eutrophic and hypertrophic species in community; (3) finally, the Moszczenica river below the reservoir was characterized by eutrophic to hypertrophic conditions and poor ecological status, with 68% participation of eutrophic and hypertrophic species in the community. During the cyanobacterial bloom, no diatoms were observed in the phytoplankton. Beginning from October 2019, mainly the presence of *Aulacoseira granulata* (84.7%), *Stephanodiscus hantzschii* (10.7%), and single valves of *Cyclostephanos dubius* and *Nitzschia supralitorea* were recorded in samples.

The bioassessment of the river and reservoir confirms the hydrochemical results. Both the river supplying the artificial reservoir and the water basin itself are heavily loaded with biogenic compounds; as such, the trophic conditions of ecosystems before the summer season only favor the development of algal blooms.

### 3.2. Investigation of Cyanobacterial Bloom

In 2019, the first visual signs of the cyanobacterial bloom in the Stryków artificial reservoir were noticed on 17 June. The bloom was then maintained until the end of October 2019 (Figure 5).

The microscopic analysis of the samples showed that the cyanobacteria in the phytoplankton were present in all samples collected during the visible bloom (*n* = 9). Twelve cyanobacteria taxa were identified in the samples (Figure 6), with the highest taxonomic diversity in the *Microcystis* genus. The constant species were the following: *M. aeruginosa* (Kützing) Kützing, *M. wesenbergii* (Komárek) Komárek ex Komárek, *M. viridis* (Braun) Lemmermann, and *M. flos-aquae* (Wittrock) Kirchner. Some dynamics were observed in relation to *Aphanizomenon flos-aquae* Ralfs ex Bornet and Flahault and *Dolichospermum crassum* (Lemmermann) Wacklin, Hoffmann and Komárek. The rest occurred either at the beginning of the season—*M. novacekii* (Komárek) Compère, *D. smithii* (Komárek) Wacklin, Hoffmann, and Komárek and *Anabaenopsis* sp., or at the end of it—*M. ichthyoblabe* (Kunze) Kützing, *D. flos-aquae* (Brébisson ex Bornet and Flahault) Wacklin, Hoffmann, and Komárek.

The quantitative analysis of the cyanobacteria expressed as the biomass of taxa revealed some dynamics in the phytoplankton communities (Figure 7). In each sample, the genus *Microcystis* dominated, with a variable percentage share of particular taxa, while the remaining cyanobacteria only accompanied in the bloom, with a total share of up to 6.4%. The highest percentage in the samples was recorded for *M. viridis* biomass, max 73.4% for sample S050719, followed by 68.1% for S150919, and 62.2% for S220819. At the beginning of bloom, the following high percentage of biomass was also observed for *Microcystis* spp.: 50.2% for S210619 and 42.9% for S170619, while some peaks were observed for *M. ichthyoblabe* (44.9%—S080819), *M. wesenbergii* (42.4%—S220719), *M. flos-aquae* (38.9%—S191019) and *M. aeruginosa* (24.3%—S080819) biomass.

In each sample from the reservoir where cyanobacteria were recorded in phytoplankton, the total biomass of taxa exceeded 3 × 10^6^ μm^3^/mL; therefore, the cyanobacteria formed a bloom from June to October (Figure 8). The highest cyanobacterial biomass (39.14 × 10^6^ μm^3^/mL) was on 21 June, and the lowest (3.28 × 10^6^ μm^3^/mL) on 19 October. In the second half of July and at the turn of August–September, an increase in biomass was observed compared to the previously-collected samples.

### 3.3. Multivariate Data Analysis

Hydrochemical factors (See: Section 3.1; Table 1) explained 92.3% of the variation in cyanobacteria composition. The interactive-forward-selection test indicated that water temperature had the greatest influence on the variation of taxa in communities, explaining 18.4% of the variation, while EC accounted for 13.4%, N/P 13.3%, P-PO_4_ 10.5%, pH 10.2%, N-NO_3_ 9.9%, Dissolved O_2_ 8.5%, N-NH_4_ 8.1%. Factors accounting for <10% were excluded from the PCA analysis.

The two principal components with the highest explanation (W-Temp and EC) showed a positive mutual correlation, and the analysis placed them in the first quarter of ordination space. The PCA indicated that both factors caused an increase in *Aphanizomenon flos-aquae* biomass in communities and positively correlated with the development of all *Dolichospermum* taxa (Figure 9A; circle I). Moreover, the analysis grouped all taxa from the *Microcystis* genus in one-quarter of ordination space (IV quarter) together with N/P ratio (Figure 9A; circle II). High biomass, one exceeding three times the biomass threshold value for a bloom, was recorded in samples positively correlated with W-Temp, EC, and the N/P ratio (Figure 9B). The biogenic compounds, P-PO_4_, and the ratio of N-NO_3_ to P-PO_4_ explained together 23.8% of the variation; in addition, they demonstrated a positive correlation with the presence of cyanobacteria for the *Microcystis* group and slightly negative for the *Dolichospermum* group. Moreover, the analysis indicated that an increasing concentration of P-PO_4_ itself did not correlate with any increase in cyanobacterial biomass in communities (Figure 9B).

### 3.4. Investigation of Cyanobacterial Toxins

All three types of cyanobacterial toxins were already present in the first sample collected at the beginning of the bloom (Figure 10) and remained present in all samples taken until 11 July, when cylindrospermopsin was not recorded. In all samples, microcystin LR was the dominant toxin, whose concentration significantly exceeded that of the others. The total concentration of cyanobacterial toxins in the water was lower in June than in July. Their concentration also gradually increased over time, together with a decrease in total cyanobacterial biomass, reaching more than twice the value in the last sample compared to the previous ones.

## 4. Discussion

With the progression of climate change, the occurrence of cyanobacterial blooms has become a growing phenomenon on a global scale. Freshwater blooms not only pose a great threat to public health due to their toxic secondary metabolite production, but also exert a range of adverse negative environmental impacts [8]. Climate change, and its resulting increase in mean air and water temperature, as well as in patterns of water precipitation, favors the development of cyanobacteria in ecosystems [18,36]. Predictions of CyanoHAB based on screening surveys in the United States show an increasing trend in bloom prevalence from 7 days per ecosystem in current conditions to 18–39 days in 2090 [37]. The mean temperatures of the area where the Stryków reservoir was constructed have increased by 1.5 °C over the past 20 years, which has prompted the formation and persistence of a massive cyanobacterial bloom. The first visual signs of cyanobacterial expansion appeared within days of the start of the sampling period, and due to their high growth rate, they soon dominated the other organisms in the planktonic community for an extremely long period—125 days. As well as being photoautotrophic, cyanobacteria are known to employ a range of ecophysiological strategies to outcompete eukaryotic algae and ensure their mass dominance in phytoplankton, such as the presence of gas vesicle aggregates, rapid cell division, the use of specific metabolic pathways, making them independent of aqueous forms of nitrogen, and finally the ability to accumulate P and trace metals [38,39,40].

Global warming also exerts an indirect effect on CyanoHAB expansion through its influence on eutrophication, the most important driver of blooming. Reservoirs are very susceptible to climate change. The combination of increased air temperature with decreased precipitation, which was clearly noticeable in the studied area, together with an increase in water temperature, intensifies evaporation, thus increasing pollutant concentration in aquatic ecosystems [41]. Eutrophication is also driven by anthropogenic sources, most commonly the allochtonous input of nutrients to freshwater ecosystems due to surface runoff from agricultural areas or the direct discharge of municipal/industrial wastewaters [42]. However, in the case of flow-through artificial reservoirs, the river supplying the water basin itself is also a significant source of nutrients. River-fed lakes and reservoirs function as collectors for all the unwanted components carried by the river flow [43]; they may also serve as tools supporting the auto-purification process of the river [41].

The Moszczenica river has always carried high nutrient loads to the reservoir. As the upper part of the river was classified as eutrophic, the Stryków reservoir and the river below the reservoir were hypertrophic, carrying high total phosphorus loads downstream. In addition, high levels of nutrient accumulation were favored by the shallow depth of the constructed reservoir and the fact that it was never emptied, and together with the constant water mixing after wind blows, nutrients return from sediment into the water column. This coincidence of global warming forces, ecosystem characteristics, and human impact, through its influence on river habitat and the nutrient enrichment of the catchment area, inevitably leads to the formation of a heavy and extremely long cyanobacterial bloom.

CyanoHABs threaten the following ecological stability and integrity of ecosystems: they decrease water clarity and can locally increase surface water temperature [44]; in addition, when dying, the blooms increase the amount of organic matter, whose decomposition leads to anoxia and hypoxia [13]. In the first half of July 2019, the waters of the Stryków reservoir were characterized by a lower oxygen content than before the bloom, and this was clearly related to a high drop in cyanobacterial biomass in phytoplankton. The aerators set at the reservoir were insufficient to improve oxygenation. While the causes of the bloom collapse in autumn are quite easy to explain, i.e., a decrease in the air and water temperature, such fluctuations in the middle of the summer season are more interesting. Since no human activity has been made to eliminate the bloom, there is a high possibility that it may have been limited by cyanophages or predatory bacteria secreting lysing agents [45,46]. There are some evidences that these predators increase their activity as the water in ecosystems gets warmer [47], and the 30 June was characterized by the highest air temperature over the year 2019. Unfortunately, this decrease in cyanobacterial biomass was accompanied by an increase in cyanotoxin content in the water. While it is widely known that large amounts of toxins are released into the water during cell death, this release can also be forced by environmentally mediated circumstances, such as relatively sudden nutrient limitations [48]. Such a strong decrease was observed in the Stryków reservoir between 17 and 21 June, when the ratio of nitrates to phosphates dropped from 32.7 to 0.2.

Our observations of the nutrient concentration in the water supplying the reservoir in the spring season suggest that it may be used to predict not only the occurrence of a bloom but also the composition of cyanobacteria among the phytoplankton. As phosphorus has traditionally been considered the major factor in algal blooming, nitrogen inputs are less important, especially in the case of N_2_-fixing cyanobacteria [13]. However, the key role in shaping the composition of cyanobacteria in the bloom seems to be played by the ratio of nitrogen to phosphorus as follows: in the present study, N/P explained 13.3% of the variation in cyanobacterial composition, while P-PO_4_ explained 10.5%. Non-N_2_-fixing genera, such as *Microcystis*, require the presence of nitrates in water; as such, if the ecosystem services provide this nutrient in excess, it promotes their massive development. However, phosphorus enrichment in ecosystems with low nitrogen levels favors the development of N_2_-fixing genera, which fulfill their nitrogen demands by converting atmospheric N_2_ [49]. The Stryków reservoir was dominated by the *Microcystis* genus, and the N/P ratio was the major environmental factor responsible for its persistence. In the community, *M. viridis* and *M. wesenbergii* dominated and outnumbered *M. aeruginosa* in at least two samples. In a study of *Microcystis* growth rates, Yamamoto and Tsukada [50] indicate small differences between *M. viridis* and *M. wesenbergii* (maximum 0.63 d^−1^ and 0.61 d^−1^, respectively) and a much lower rate for *M. aeruginosa* (max 0.38 d^−1^), which provides some insights into the structure of the cyanobacterial bloom in the Stryków artificial reservoir. Among the N_2_-fixing genera, the *Aphanizomenon* and *Dolichospermum* presence was positively correlated with the high temperature and electric conductivity of water. While investigating the environmental factors of cyanobacterial dominance, Kim and co-authors [51] found that EC was the major variable in shaping the cyano-biomass. Moreover, as EC can be easily monitored with sensors, they assume it can be used as an important variable for the prediction of algal blooms. The species’ dominance in the environment never results from the impact of only one factor—it is determined according to the complex nonlinear relationships involving many variables. Nevertheless, nutrient over-enrichment of water bodies undoubtedly promotes the growth of CyanoHABs. The total biomass of cyanobacteria from the studied reservoir was influenced by the N/P ratio as follows: The total biomass increased in the presence of a high ratio and fell with the depletion of nutrient resources; however, the decomposition of dead cells, the constant nutrient supply from the river inflow, and the accompanying high water temperature (>20 °C) induced biomass regrowth twice, but never to the amount observed at the beginning of the phenomenon. These observations confirm that the eutrophic river supplying the reservoir plays an important role in inducing nutrient accumulation during spring and accelerating the bloom when the weather conditions favor the development of cyanobacteria.

This detailed case study of a long-lasting cyanobacterial bloom in an artificial, flow-through recreational reservoir demonstrates the important role played by the combination of natural forces, such as climate change and the environmental alterations caused by human activity. In this case, these factors were exacerbated by the pressure exerted by the transformed, nutrient-enriched river, thus increasing the chance of CyanoHABs.

## 5. Conclusions

A combined analysis based on hydrochemical analysis of river and reservoir water before the growing season, weather forecasting, and the determination of the trophic levels of the aquatic ecosystem may form a key element in predicting the occurrence of cyanobacterial blooms and their taxonomical structure in the summer season. Our case study highlights the issue of improper ecosystem management. It is inevitable that reservoirs whose catchment is subjected to long-term anthropogenic effects (agricultural use in this case) and which receive a constant inflow of large biogenic loads from their catchment will suffer from the development of potentially toxic blooms, which will limit the possibility of the reservoir being used for recreational purposes. This process is accelerated by global climate change, resulting in the presence of a persistent bloom, representing a danger to human health and preventing the use of the reservoir for its intended purpose.

## Figures and Tables

**Figure 1 ijerph-19-06990-f001:**
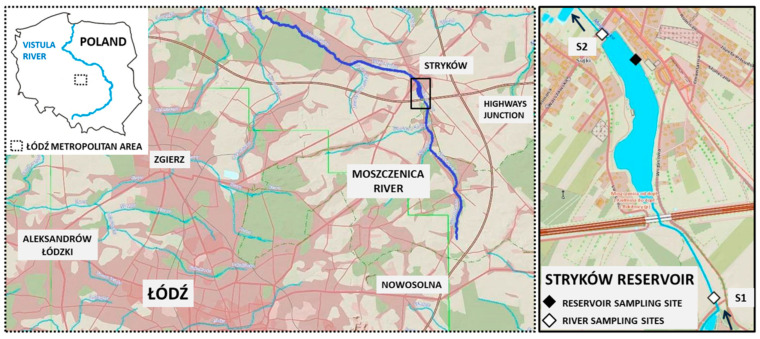
Location of the Stryków artificial reservoir and the study sampling sites; arrows indicate direction of river flow.

**Figure 2 ijerph-19-06990-f002:**
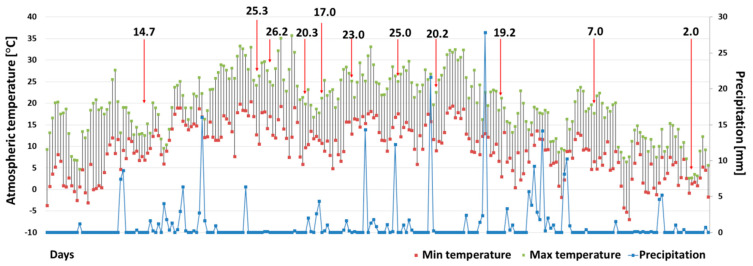
The range of daily air temperatures and the daily sum of precipitation in the research season of 2019 (1 April 2019–30 November 2019). The arrow indicates the air temperature on the sampling day.

**Figure 3 ijerph-19-06990-f003:**
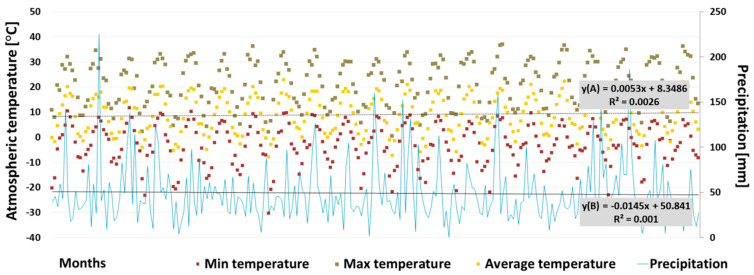
The range of monthly air temperature and monthly sum of precipitation over 20 years (January 1999–December 2019); y(A)—trend line of mean monthly air temperature; y(B) trend line of precipitation.

**Figure 4 ijerph-19-06990-f004:**
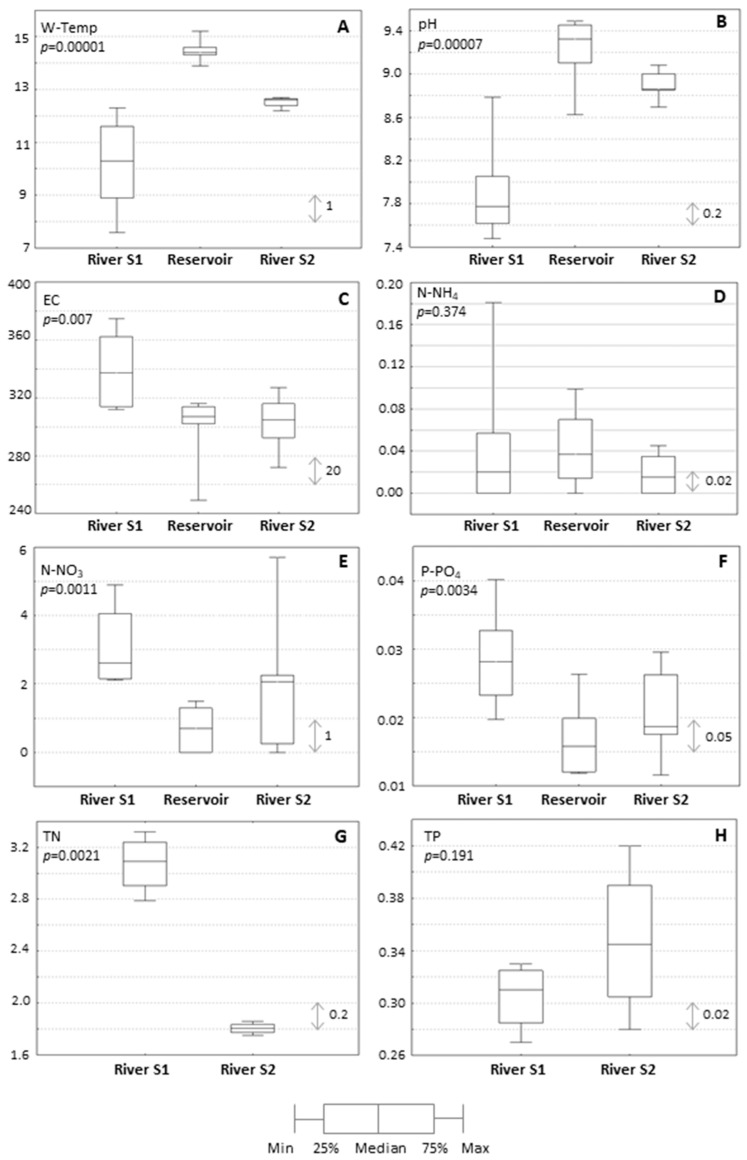
The hydrochemical parameters of the Moszczenica river and Stryków reservoir; graphs express diurnal measurements preceding the formation of heavy bloom; (**A**): water temperature [°C], *p* < 0.05; (**B**): water pH, *p* < 0.05; (**C**): electric conductivity [μS/cm], *p* < 0.05; (**D**): ammonium [mg/L], *p* > 0.05; (**E**): nitrates [mg/L], *p* < 0.05; (**F**): ortophosphates [mg/L], *p* < 0.05; (**G**): total nitrogen [mg/L], *p* < 0.05; (**H**): total phosphorus [mg/L], *p* > 0.05.

**Figure 5 ijerph-19-06990-f005:**
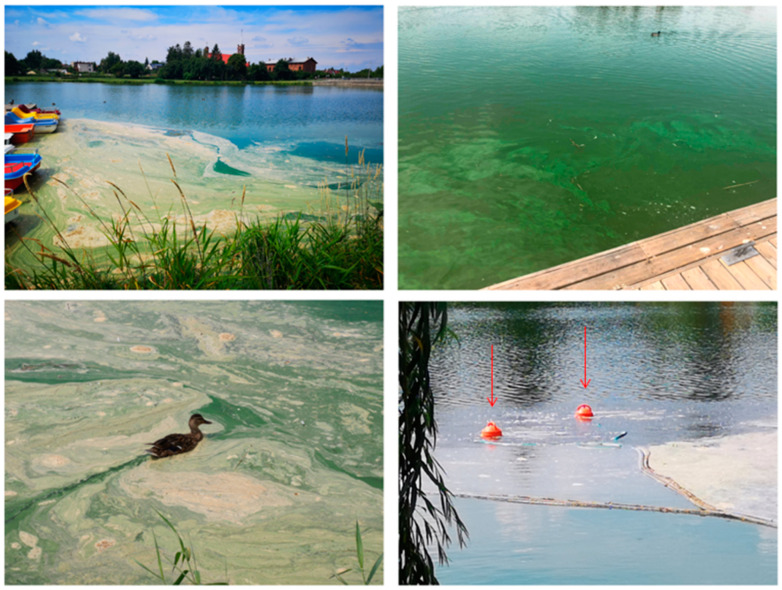
The cyanobacterial bloom in Stryków reservoir; arrows point at the aerators located at the surface, photo from 17 June 2019.

**Figure 6 ijerph-19-06990-f006:**
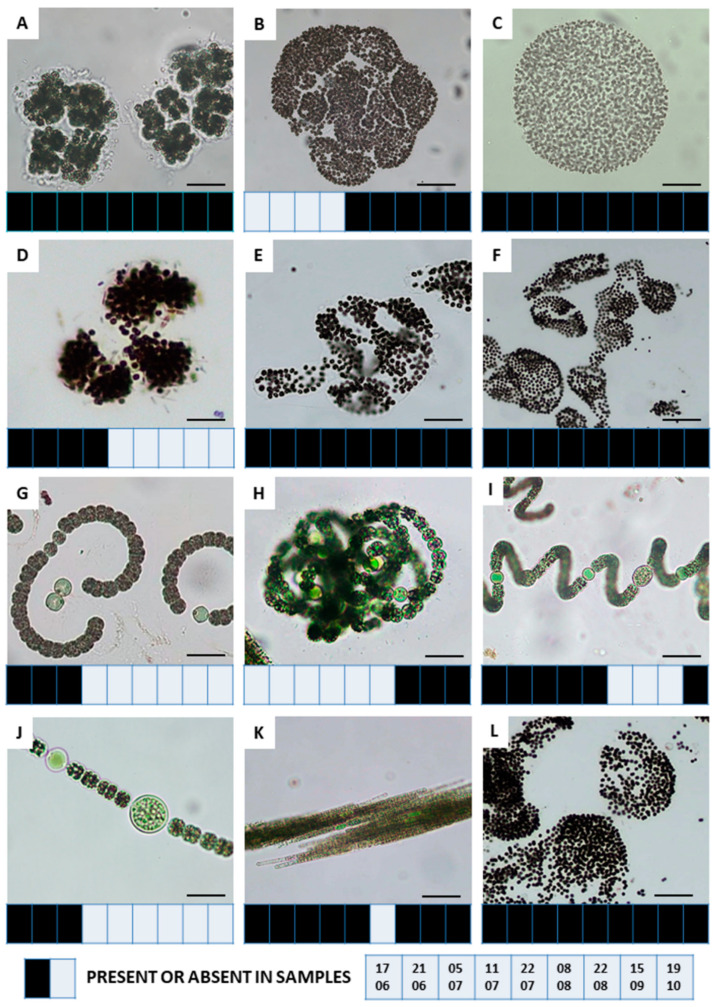
The cyanobacteria taxa and their presence in samples from the Stryków artificial reservoir; (**A**): *Microcystis viridis* (scale bar 20 μm), 21 June 2019; (**B**): *M. ichthyoblabe* (scale bar 40 μm), 22 August 2019; (**C**): *M. flos-aquae* (scale bar 30 μm), 11 July 2019; (**D**): *M. novacekii* (scale bar 15 μm), 21 June 2019; (**E**): *M. wesenbergii* (scale bar 20 μm), 21 June 2019; (**F**): *M. aeruginosa* (scale bar 40 μm), 21 June 2019; (**G**): *Anabaenopsis* sp. (scale bar 20 μm), 5 July 2019; (**H**): *Dolichospermum flos-aquae* (scale bar 15 μm), 15 September 2019; (**I**): *D. crassum* (scale bar 20 μm), 22 July 2019; (**J**): *D. smithii* (scale bar 20 μm) 21 June 2019; (**K**): *Aphanizomenon flos-aquae* (scale bar 30 μm), 5 July 2019; (**L**): *Microcystis* spp. (scale bar 30 μm), 21 June 2019.

**Figure 7 ijerph-19-06990-f007:**
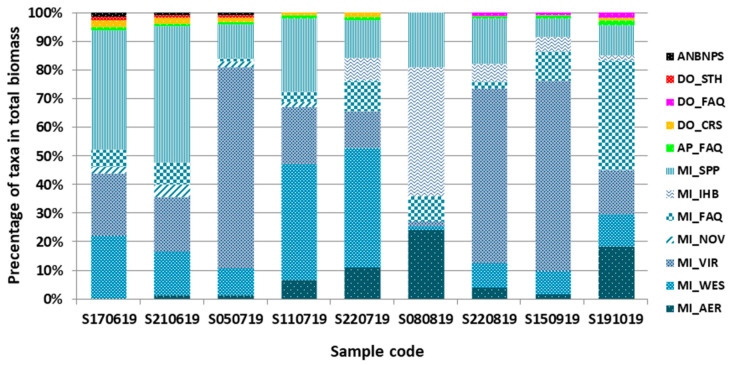
Qualitative and quantitative structure of cyanobacterial bloom in the samples from Stryków artificial reservoir based on the percentage share of taxa in the total biomass; sample code consist of: S (Stryków reservoir), and date of sampling; code of taxa: ANBNPS—*Anabaenopsis* sp., DO_STH—*Dolichospermus smithii*, DO_FAQ—*D. flos-aquae*, DO_CRS—*D. crassum*, AP_FAQ—*Aphanizomenon flos-aquae*, MI_SPP—*Microcystis* spp., MI_IHB—*M. ichthyoblabe*, MI_FAQ—*M. flos-aquae*, MI_NOV—*M. novacekii*, MI_VIR—*M. viridis*, MI_WES—*M. wesenbergii*, and MI_AER—*M. aeruginosa*.

**Figure 8 ijerph-19-06990-f008:**
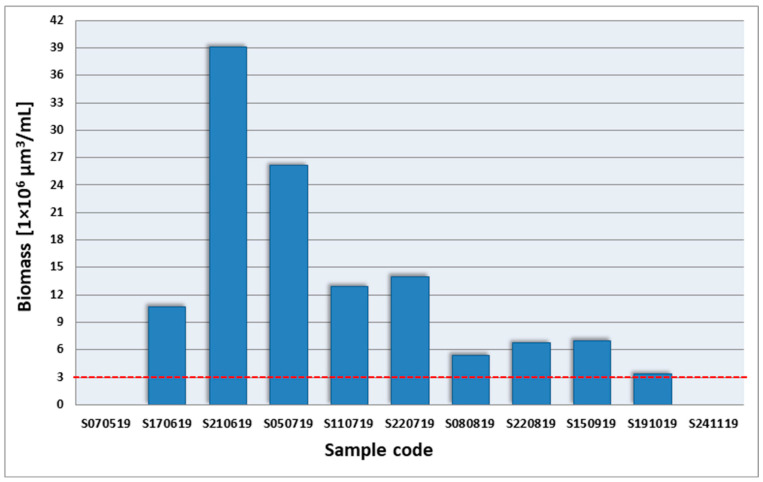
Total biomass of identified cyanobacteria taxa in the samples from Stryków artificial reservoir; red line is the threshold biomass value for bloom phenomenon.

**Figure 9 ijerph-19-06990-f009:**
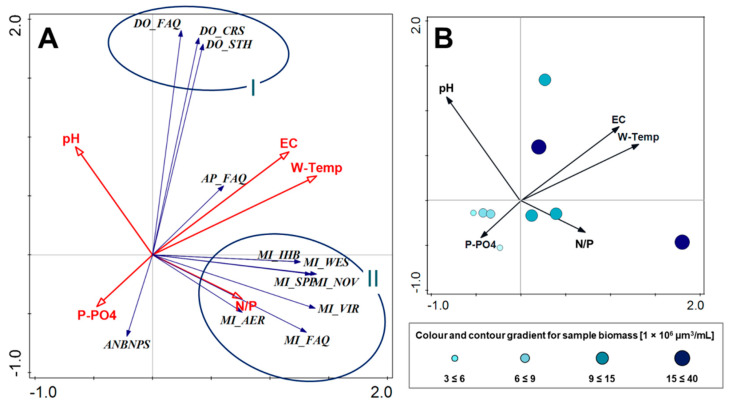
The PCA ordination plot displaying the relations between the biomass of cyanobacteria and hydrochemical factors; (**A**): relations between cyanobacteria taxa and studied factors; (**B**): relations between total cyanobacteria biomass in samples and studied factors.

**Figure 10 ijerph-19-06990-f010:**
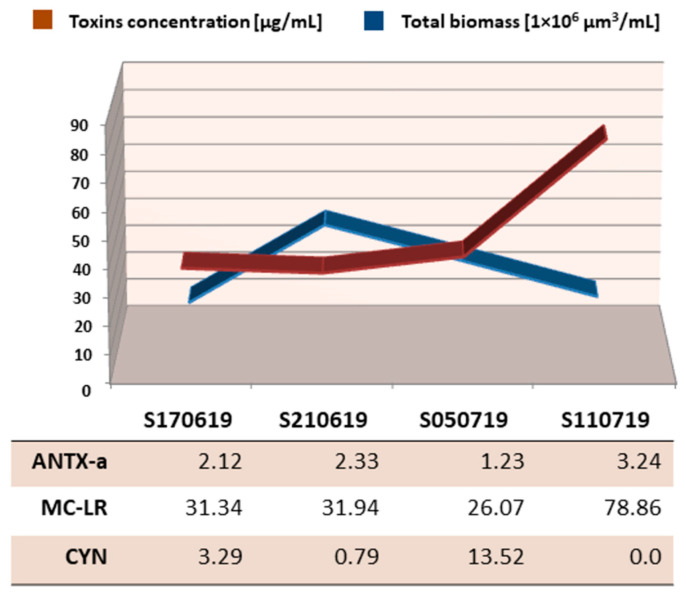
Concentration of cyanotoxins in samples from Stryków artificial reservoir against total cyanobacterial biomass.

**Table 1 ijerph-19-06990-t001:** The physical and chemical parameters of water in Stryków artificial reservoir during the research season; sample code consist of: S (Stryków reservoir), and date of sampling.

Sample Code	W-Temp [°C]	pH	Diss-O_2_ [mg/L]	EC [μS/cm]	N-NH_4_ [mg/L]	N-NO_3_ [mg/L]	P-PO_4_ [mg/L]	N-NO_3_ /P-PO_4_
S070519	14.6	9.1	15.30	305	0.02	0.64	0.02	32.0
S170619	24.6	8.4	16.50	313	0.05	0.98	0.03	32.7
S210619	24.6	9.1	16.97	296	0.02	0.01	0.06	0.2
S050719	20.4	8.8	10.45	287	0.01	0.28	0.16	1.8
S110719	17.1	9.1	13.22	287	0.02	0.30	0.43	0.7
S220719	21.7	8.8	16.67	252	0.04	0.29	0.11	2.6
S080819	22.3	9.0	15.73	234	0.02	0.38	0.14	2.7
S220819	20.8	9.0	13.25	220	0.01	0.24	0.03	8.0
S150919	19.1	8.5	13.11	209	0.01	0.41	0.01	41.0
S191019	9.7	8.3	12.64	222	0.01	0.41	0.22	1.9
S241119	5.0	7.8	10.47	227	0.05	1.26	0.09	13.8

## Data Availability

All data generated during this research were presented in the manuscript text, figures, and tables.

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
