# Peer review of "Persistent Cyanobacteria Blooms in Artificial Water Bodies—An Effect of Environmental Conditions or the Result of Anthropogenic Change"

_ijerph, 2022, doi:10.3390/ijerph19126990_

Round 1

Reviewer 1 Report

The manuscript “Persistent Cyanobacteria Blooms In Artificial Water Bodies – An Effect Of Environmental Conditions Or The Result of Anthropogenic Change” is well written with good English. It describes seasonal changes in nutrients compositions and hydrochemical parameters of Moszczenica river and Stryków reservoir (Poland) influencing growth and composition of cyanobacterial water bloom. The data has more local significance and limited novelty. On the other hand, cyanobacterial species observed are common in Europe and the results may can help generalize the knowledge among European localities with cyanoHABs.

Author Response

Dear Reviewer 1

Thank you for this pleasant opinion. Every large-scale studies need a small-scale driver and especially to understand cyanoHABs phenomenon, all local scale studies may be important, bringing some footsteps for a greater view.

With kind regards Paulina Nowicka-Krawczyk and co-authors

Reviewer 2 Report

Excellent work.  Well written and well supported presentation of the data.

Author Response

Dear Reviewer 2

Thank you for this pleasant opinion. We have done some minor English spelling correction and supported the text according to reviewers suggestion.

With kind regards

Paulina Nowicka-Krawczyk and co-authors

Reviewer 3 Report

Manuscript: Persistent Cyanobacteria Blooms In Artificial Water Bodies – An Effect Of Environmental Conditions Or The Result Of Anthropogenic Change.

The authors documented the algal abundance, composition, physiological-chemical parameters, and cyanotoxins in a Polish artificial reservoir (Stryków artificial reservoir), and found that temperature and electronic conductivity are closely associated with the Aphanizomenon biomass whereas the N/P ratio is associated with the Microcystis biomass. Generally, studying the occurrence of harmful cyanobacterial blooms and its association with environmental factors can be interesting and important. However, for publication, the present manuscript requires the major revision, especially with focus on the abstract and discussion section. Apparent, there is the lack of clear focus about the major finding. The reviewer suggests that the major findings of this study should be clearly itemized in both abstract and discussion sections. In the discussion, the meaning of the present should be also discussed.

Several different cyanotoxins such as MC-LR, anatoxin-a were simultaneously measured in this work by using the HPLC method.  Is this method newly developed in this work, or it was based on the published one? What are the detection limitations of these cyanotoxins in this method? Theses should be clearly mentioned.

Thirdly, did the authors look into the association between various cyanotoxin level and the different cyanobacterial species? It will be interesting and important to establish their link if possible.

Additionally, the following minor points should be also addressed.

  1. line 50: “Anabaena) Cylindrospermopsis, Anabaenopsis, Planktothrix, and Nodularia [12,13].”. The comma should be added between the right bracket and “Cylindrospermopsis”.
  2. Line 70: “The global increase in mean temperature has been accompanied by an increase in .” it should read like “ the global climate warming has been accompanied by an increase…”
  3. Line 73: “gain a quantitative advantage over other algae in phytoplankton blooms”. The word quantitative is not necessary.
  4. Figure 4: the statistical analysis should be conducted to check whether there is the significant different among groups, additionally, the box and whiskers on the box plot should be also detailed, for example, which is the median line, 25% percentile etc.?
  5. Figure 5. The date information for taking the picture of the reservoir should be provided here.
  6. Figure 6, the date information for taking the picture of the cyanobacterial samples should be mentioned here.
  7. Figure 7: the full name of each abbreviation in the right section of the figure such as MI_AER should be provided in the figure caption.
  8. Figure 7: the title of y axis should be date of the sampling, other than the samples. From the sample name e.g. S170619, the reviewer speculates this sample was collected on June 17, 2019. Additionally, the title of y axis should be named here, it can be the percentage of algal species.
  9. Which software was used to conduct the PCA analysis, this should be mentioned in the text.

Author Response

Dear Reviewer 3

Thank you for all your comments and suggestions which were valuable and constructive. Below, we would like to address the issues you raised in revision.

1. There is the lack of clear focus about the major finding. The reviewer suggests that the major findings of this study should be clearly itemized in both abstract and discussion sections

Thank you for this comment. We have added the major findings in abstract and addressed findings to additional recent reference in discussion – https://doi.org/10.3390/w11061163.

2. Several different cyanotoxins such as MC-LR, anatoxin-a were simultaneously measured in this work by using the HPLC method. Is this method newly developed in this work, or it was based on the published one? What are the detection limitations of these cyanotoxins in this method?

This method was previously used in a study – proper reference was added to the method section (https://doi.org/10.3390/cells10030699) and as the reviewer suggested we have supplemented the section with toxins limits of detection.

3. Thirdly, did the authors look into the association between various cyanotoxin level and the different cyanobacterial species? It will be interesting and important to establish their link if possible.

After this question we have done simple analysis in Canoco software to visualize the association (Fig. 1_attachment to response). The presence of CYN was positively related to M. viridis biomass, MC-LR to M. aeruginosa, while ANTX-a to M. wesenbergii.

However, after consideration we have decided not to support the manuscript with these data. For clarification we would like to point out that:

(a) in environment where the structure of bloom is formed by several toxin producing taxa and most of them have the ability to produce toxic compounds (i.e. despite Microcystis, the active MC-synthase genes were detected in Dolichospermum and Aphanizomenon) it is hardly to indicate which strain is responsible for the production of the exact toxin;

(b) as the toxicity of Microcystis wesenbergii has been many times debated, some reports clearly indicate that it is non-toxic – https://doi.org/10.1128/aem.55.12.3202-3207.1989, while others bring evidence for its toxicity, since it poses the mcyE toxigenic sequences https://doi.org/10.3390/app11010357; here in our study we cannot be sure on the abilities of the strains from Stryków reservoir;

(c) in this case the amount of data on this interesting association does not allow for diving any scientific-based conclusions. In this study we only screened the water for the presence of toxins at the beginning of bloom to know if even from early beginning we have an emerging issue of an environmental threat posed by the presence of cyano-toxins.

4. The following minor points should be also addressed: line 50 - lack of comma; Line 70 and 73 – style correction; Fig 5 and 6 date of taking pictures; Fig. 7 the abbreviation of taxa codes and axes to improve.

We have done all improvements according to reviewer suggestion, in case of axis x label (Fig. 7 and 8), we have decided to unify the label across the manuscript and changed to “sample code”.

5. Figure 4: the statistical analysis should be conducted to check whether there is the significant different among groups, additionally, the box and whiskers on the box plot should be also detailed, for example, which is the median line, 25% percentile.

We have supported the figure and figure caption with the significance level and included the legend for box and whiskers. We do agree that such data from the statistical point of view is important, therefore we have added this results. On the other hand, as an environmental biologists and ecologists we do not absolutely rely on p values. There are some microorganisms highly sensitive toward certain factors, that even a minor change – not statistically important may alter the qualitative and quantitative structure of community.

6. Which software was used to conduct the PCA analysis, this should be mentioned in the text.

This was already included in the paragraph – it was Canoco for Windows software.

With kind regards,

Paulina Nowicka-Krawczyk and co-authors

Round 2

Reviewer 3 Report

The reviewer has no further major concerns.